# Integrated analysis of RNA-seq datasets reveals novel targets and regulators of COVID-19 severity

Thais Teixeira Oliveira , Júlia Firme Freitas , Viviane Priscila Barros de Medeiros, Thiago Jesus da Silva Xavier, Lucymara Fassarella Agnez-Lima

**During the COVID-19 pandemic, RNA-seq datasets were produced to investigate the virus–host relationship. However, much of these data remains underexplored. To improve the search for molecular targets and biomarkers, we performed an integrated analysis of multiple RNA-seq datasets, expanding the cohort and including patients from different countries, encompassing severe and mild COVID-19 patients. Our analysis revealed that severe COVID-19 patients exhibit overexpression of genes coding for proteins of extracellular exosomes, endomembrane system, and neutrophil granules (e.g., *S100A9*, *LY96*, and *RAB1B*), which may play an essential role in the cellular response to infection. Concurrently, these patients exhibit down-regulation of genes encoding components of the T cell receptor complex and nucleolus, including *TP53*, *IL2RB*, and *NCL*. Finally, SPI1 may emerge as a central transcriptional factor associated with the up-regulated genes, whereas TP53, MYC, and MAX were associated with the down-regulated genes during COVID-19. This study identified targets and transcriptional factors, lighting on the molecular pathophysiology of syndrome coronavirus 2 infection.**

## Introduction

The severe acute respiratory syndrome coronavirus 2 (SARS-CoV-2) has triggered the largest pandemic in recent history, with more than 6.9 million confirmed deaths identified as the etiologic agent of Coronavirus Disease 2019 (COVID-19) (World Health Organization, <https://covid19.who.int>. Accessed on 03 July 2023), before SARS-CoV-2, two other epidemics were caused by betacoronaviruses. The severe acute respiratory SARS-CoV in 2002 and the Middle East respiratory syndrome in 2012 demonstrated the potential of this virus family for human-to-human transmission. The recurrence of these viruses makes understanding their pathogenesis essential (Chen et al, 2020b).

COVID-19 primarily affects the respiratory system but also leads to other symptoms, including taste and olfactory disturbances, gastrointestinal issues, neurological complications, cardiovascular problems, ocular symptoms, coagulation disorders, or even present as asymptomatic (Baj et al, 2020; Giannis et al, 2020; Wang et al, 2020). The COVID-19 disease is categorized into four groups: mild, moderate, severe, and critical, with the critical manifestation characterized by acute respiratory distress syndrome (ARDS) and sepsis or septic shock (Yang et al, 2020; Zhou et al, 2020). Furthermore, several risk factors have been associated with the disease, including advanced age, male sex, overweight, hypertension, diabetes, and preexisting respiratory and cardiovascular conditions (Zhou et al, 2020; Hu et al, 2021).

SARS-CoV-2 requires the host's cellular apparatus for replication, assembly, and egress from cells. Because of this, the COVID-19 disease is associated with dysregulation of several genes and signaling pathways (V'kovski et al, 2021). The pathogenic mechanism of SARS-CoV-2 involves dysregulation of the renin-angiotensin-aldosterone system, hypoxia, hyperinflammation, cytokine and chemokines storms, and sepsis, resulting in a dysfunctional inflammatory response with persistent fevers and high levels of inflammatory markers (Tay et al, 2020). Sudden storms of cytokines and chemokines may lead to lung injury and ARDS. In addition, viral replication may result in cellular apoptosis, leading to diffuse alveolar damage and ARDS (Mason, 2020; Wu & McGoogan, 2020).

Omics technologies are instrumental in unraveling the molecular mechanisms underlying the physiological and metabolic changes in COVID-19. Despite the high cost, RNA-seq studies provide valuable information about gene expression changes in host tissues and cells after infection by SARS-CoV-2, including altered processes and pathways, favoring the understanding of pathogenesis and host–pathogen interaction (Jain et al, 2021; Yang et al, 2021; Samy et al, 2022). Transcriptomic analysis of blood samples from COVID-19 patients allows the identification of differentially expressed genes (DEGs) associated with the host's immune or inflammatory response and regulatory networks (Kwan et al, 2021; Daamen et al, 2022). Such analysis can help predict prognostic and diagnostic biomarkers and potential drug targets for pathophysiology groups, supporting personalized treatment decisions (Rodriguez et al, 2021; Iqbal & Kumar, 2022). During the 3 yr of the COVID-19 pandemic, several RNA-seq data were generated and are

Departamento de Biologia Celular e Genética, Universidade Federal do Rio Grande do Norte, UFRN, Natal, Brazil

Correspondence: lucymara.agnez@ufrn.br

publicly available for analysis. Different analyses of these data cost little and can guide future research. The integrated analysis of available data can increase the reliability of the results found and reduce bias because of the genetic variability of the participants.

In this work, we reanalyzed previously published transcriptome data from blood samples and leukocytes isolated from COVID-19 patients. We compared them with non-COVID-19 individuals to find potential biomarkers of disease severity and targets to improve the understanding of COVID-19 molecular pathogenesis. Using bio-informatics tools, we identify overexpressed genes related to the endomembrane system and neutrophil activation in hospitalized patients, and down-regulated genes are associated with the T cell receptor complex. In addition, SPI1 and other ETS family transcription factors are the predicted regulators with the most enriched binding motifs between the overexpressed genes in severe COVID-19 patients. At the same time, MAX, MYC, and TP53 were appointed as the central regulators of down-regulated genes. Our results indicate that these transcription factors are essential to gene expression control during COVID-19. These findings can potentially contribute valuable insights to understanding, diagnosing, and treating severe COVID-19.

# Results

## Data collection and description

To access genes related to the severity and progression of COVID-19, we selected four RNA-seq datasets containing public raw data from blood samples collected during the pandemic. The data were renamed according to the sample origin (e.g., whole blood was named WB01). Dataset 1, WB01 (GSE161731) samples were collected from patients in the USA and categorized into early, middle, and late disease stages based on the time from reported symptom onset. Furthermore, the work classified samples into severity categories like "Outpatients" to patients with mild COVID-19 cases not requiring hospitalization (n = 22) and "Hospitalized," representing severe cases that required hospitalization (n = 9). In this work, we selected only data with severity classification (McClain et al, 2021). Dataset 2, WB02 (GSE172114), was collected from patients admitted to the university hospital network in northeast France. A study by Carapito and colleagues previously classified the patients into "Critical" (n = 46) and "Non-critical" (n = 23) (Carapito et al, 2022). This categorization was maintained in the present work. Dataset 3, WB03 (E-MTAB-10926), was collected from patients in Spain and classified as "Mild" (n = 19), "Moderate" (n = 26), and "Severe" disease (n = 10). Moderate patients were admitted to the hospital with supportive care limited to oxygen delivery, whereas severe patients were admitted to the intensive care unit (ICU) (Jackson et al, 2022). This study combined moderate and severe patients into the severe group, as both were hospitalized.

Finally, we analyzed a dataset of leukocytes isolated from the whole blood of ICU patients (Overmyer et al, 2021). This dataset was categorized into the "COVID-19" group (ICU = 51 patients and non-ICU = 44 patients) and the "non-COVID-19" group (ICU = 16 patients and non-ICU = 10 patients). The non-COVID-19 group consisted of patients with moderate to severe respiratory issues who tested negative for SARS-CoV-2. The summary of the data classification and the workflow of this study are represented in Fig 1.

## Hospitalized patients have increased expression of genes related to transport vesicle and neutrophil activation

To analyze essential genes regulated during severe COVID-19, the DEGs in each dataset were measured, and the volcanos plot exhibits the DEGs with adjusted *P*-value < 0.05 (Fig 2A). The top 10 most significant DEGs were labeled in the graph. The up- and down-regulated genes were submitted to gene ontology (GO) analysis to identify pathways and processes vital to COVID-19 pathogenesis (Table S1). This analysis shows the enrichment of genes located in the transport vesicles, lysosomal membrane, endosome membrane, extracellular exosome, and endoplasmic reticulum (Fig 2B). However, it is worth noting that genes belonging to these components were also up-regulated in severe non-COVID-19 patients, indicating that this up-regulation is not specific to the worsening caused by SARS-CoV-2. In addition, there is an enrichment of genes located in cellular components like endoplasmic reticulum chaperone complex, clathrin-coated vesicle membrane, and endocytic vesicle only in severe COVID-19 patients' blood. In addition, other cellular components like platelets alpha granule lumen, podosome, and proteasome complex showed enrichment (Fig 2B). Up-regulated genes also were enriched in biological processes like neutrophil activation, innate immune response, actin cytoskeleton organization, and negative regulation of MAP kinase activity. On the other hand, genes related to T cell activation and adaptive immune response were down-regulated in severe COVID-19 and non-COVID-19 patients. Otherwise, genes related to the regulation of transcription, cellular response to DNA damage stimulus, tRNA aminoacylation, and maturation of SSU-rRNA were down-regulated only in COVID-19 patients (Fig 2C).

Considering that the regulation of these genes may be related to the severity of SARS-CoV-2 infection or susceptibility to it, we also assessed the master transcriptional regulators that might be associated with the expression control of these genes. Transcription factors like ELK1, ETS1, ETV6, ELF5, GABPB1, SPI1, and SPIB may have enriched binding motifs in the regulatory regions of up-regulated genes. Between down-regulated genes, transcription factors like ZNF143, ZBTB33, ATF3, and transcription factors of the STATs family have binding motifs enriched in more than one analyzed dataset (Fig 2D).

## Integrated analysis shows overexpression of genes related to exosome, lysosomal membrane, and specific granules only in severe COVID-19 patients

We constructed an upset plot to analyze common DEGs shared between the datasets. The plot shows 305 up-regulated genes that were specific to COVID-19 datasets. These genes were not up-regulated in the leukocytes of patients with negative tests for COVID-19, indicating specificity in the expression regulation of these genes. The set of 305 specific genes up-regulated only in severe COVID-19 patients was named "setUPCOVID-19." In the same way, there are 179 down-regulated genes shared in COVID-19 datasets "SetDOWNCOVID-19" (Fig 3A). Furthermore, we identified a set of 218 genes that were up-regulated in all data from patients with severe

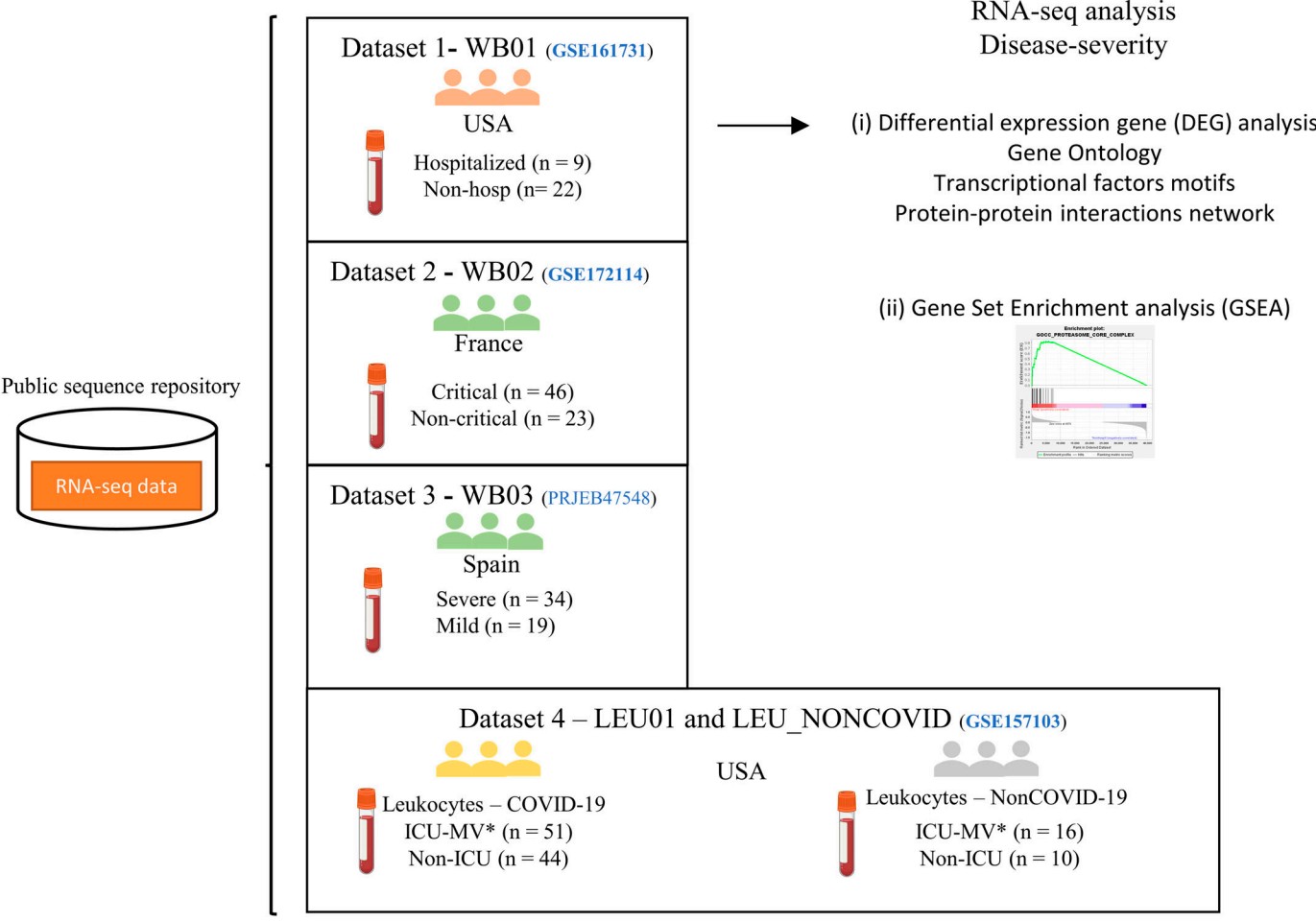

**Figure 1. Schematic representation of the systematic analysis of RNA-seq dataset.**
We analyzed 275 samples derived from blood or cells isolated from blood in COVID-19 and non-COVID-19 patients. The samples were stratified based on disease severity. The datasets were analyzed by two distinct methods: Gene set enrichment analysis and differentially expressed genes. The samples were named whole blood 01 (WB01), whole blood 02 (WB02), whole blood 03 (WB03), and leukocytes (LEU01) and NON-COVID reference. *ICU-MV—Patients in the intensive care unit who required mechanical ventilation.

respiratory syndrome "SetUPall" and 108 genes down-regulated in all severe patients "SetDOWNall." For further analysis, we focused on this set of common genes (Table S2). The 305 overexpression genes coding proteins were located in vesicle components such as extracellular exosome, azurophil granule, specific granule, and lysosomal membrane (Fig 3B). However, we observed that genes encoding vesicle and granule proteins were also enriched in non-COVID-19 patients (Fig 3C; Table S3). Although the increased expression of genes related to vesicle transport and neutrophil granules is not specific to SARS-CoV-2 infection, some genes specifically exhibited increased expression among severe COVID-19 patients, indicating these genes as targets to understand the molecular pathophysiology of the disease, and the observed differences in the severity of this disease when compared with other flu and respiratory diseases.

To evaluate the interaction network formed by the products of shared genes, the "setUPCPVID-19" and the "setUPall" were joined, and protein–protein interaction (PPI) was constructed. Resulting in a PPI network formed by 242 nodes and 399 edges (Fig 3D). Genes overexpressed in all data were represented as white circles. In contrast, the red circles represented the genes up-regulated only in severe COVID-19 patients. The centrality analysis of the PPI network shows that SPI1, GAPDH, CDC42, RAB1B, TYROBP, S100A9, and LY96 were the most critical nodes of the network with the highest values of degree and betweenness. These genes were up-regulated only in severe COVID-19 patients. Furthermore, other genes have a high centrality value in this network, such as DEFA4, NCF4, ELANE, TRIM25, FADD, and NFKBIA.

We also search for the master transcriptional regulators of the up-regulated common genes. The iRegulon analysis indicates that the ETS family of transcription factors, led by SPI1, were the regulators with the most enriched binding motifs between the up-regulated genes in severe COVID-19 patients (NES = 4.93) with 70 targets. The overexpression of SPI1 and its targets suggests that this transcription factor may be involved in COVID-19 pathogenesis or susceptibility. In addition, MAF (NES = 4.79603), NR2F2 (NES = 4.54908), NAP1L1 (4.47656), and RXRA (NES = 4.178) also can be important regulators of this network (Fig 3E). On the other

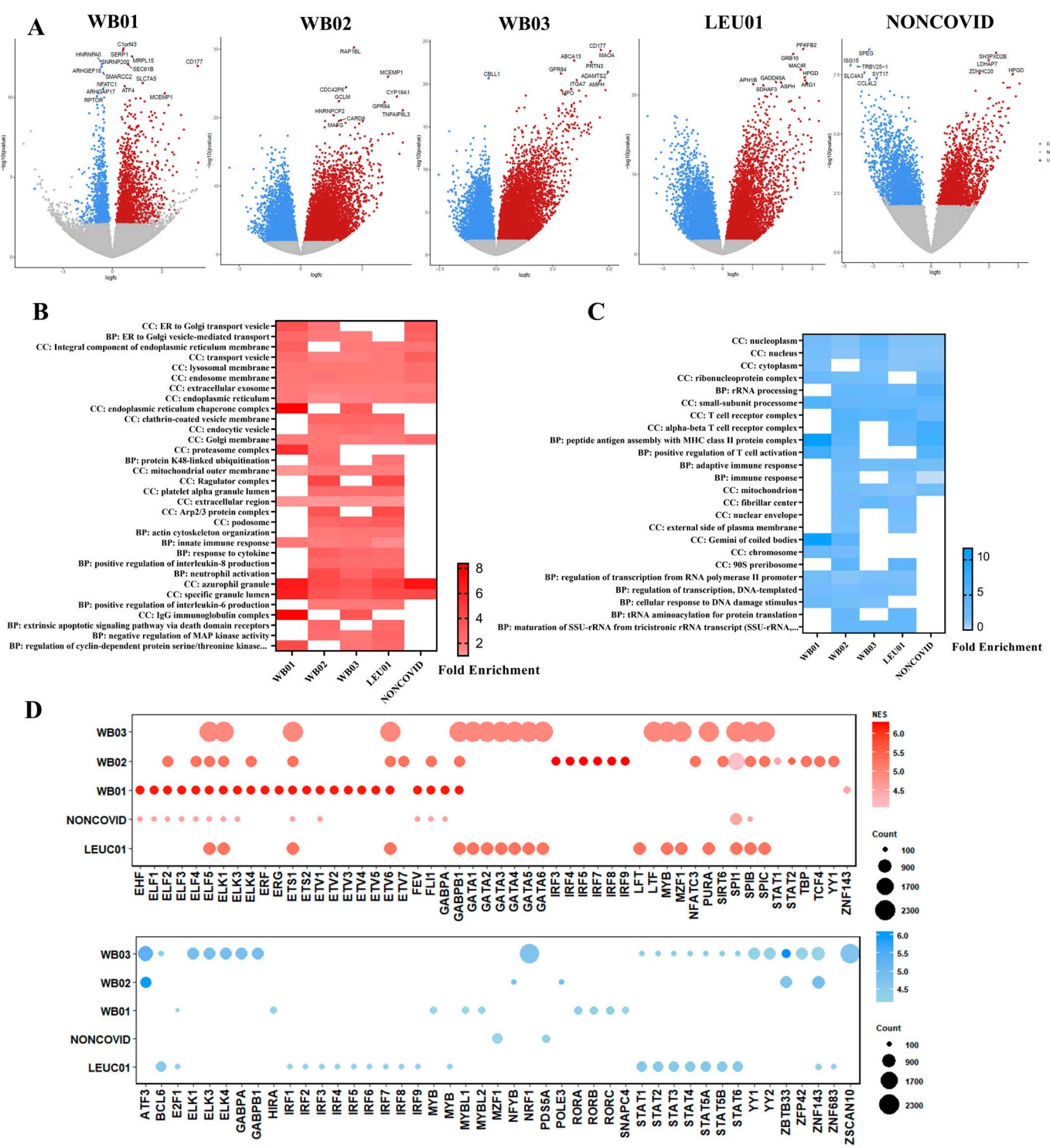

**Figure 2. Landscape of differential expression genes analysis and regulators.**
**(A)** Volcano plot of differential expression genes in analyzed data groups, colored dots show genes with adjusted *P*-value < 0.05. Blue dots represent down-regulated genes, and red dots represent up-regulated genes. **(B)** Heatmap exhibits the enriched biological process and cellular components in up-regulated genes with a false discovery rate (FDR) < 0.05. **(C)** Heatmap displaying enriched biological process and cellular components in down-regulated genes with FDR < 0.05. **(D)** Predict transcriptional regulators of up-regulated genes (above) and down-regulated genes (below). The bubble size corresponds to the number of targets related to the transcription factor. The color intensity of the bubble corresponds to the normalized enrichment score (NES). We considered NES > 4.0.

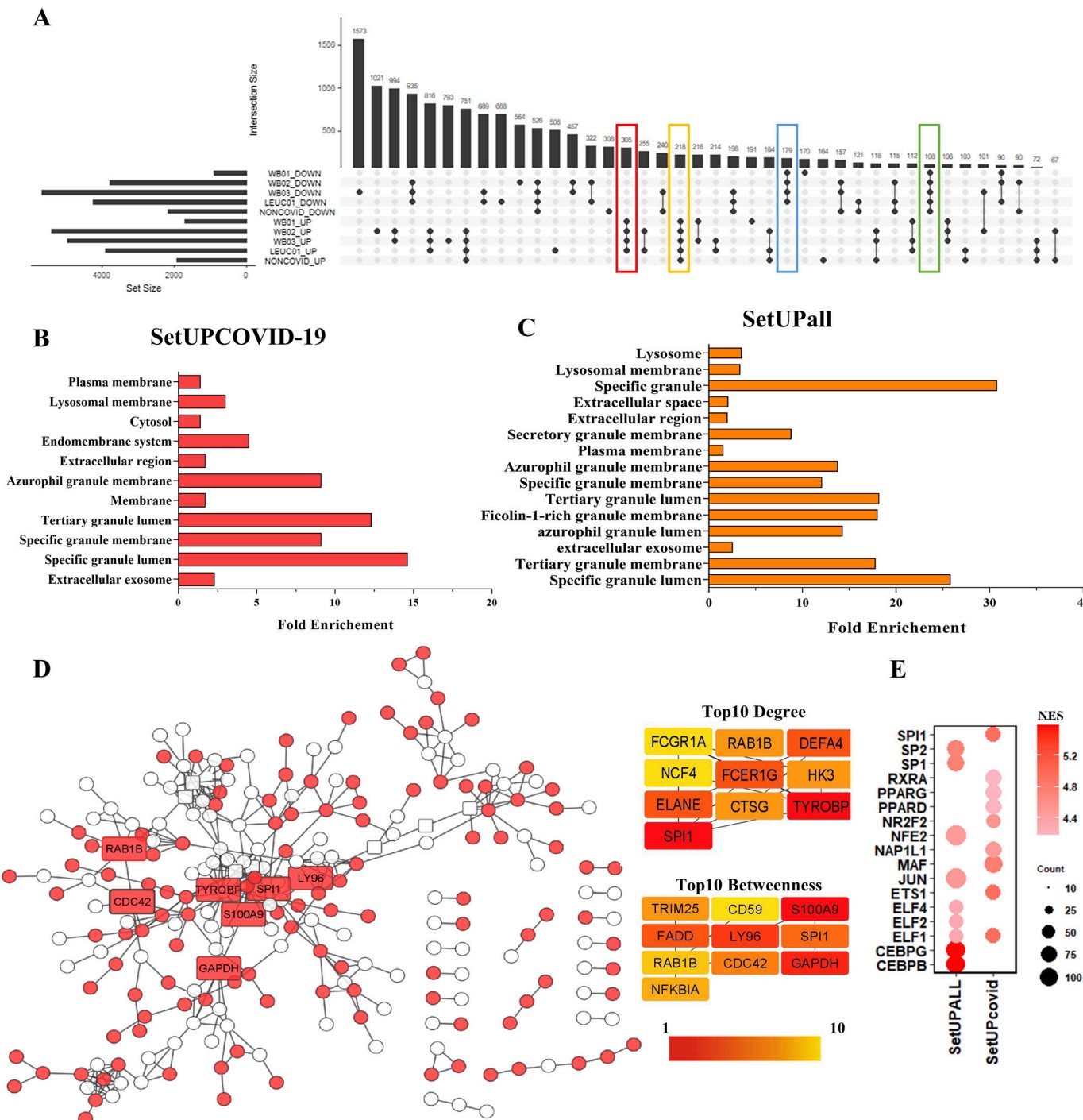

**Figure 3. SPI1, RAB1B, and S100A9 are central genes in the interaction network formed by overexpressed gene products.**

**(A)** The upset plot illustrates 305 up-regulated genes unique to severe COVID-19 patients and 218 genes up-regulated in all severe patients. Conversely, 179 genes are down-regulated in severe COVID-19 patients, and 108 genes are down-regulated in all severe patients. **(B)** Genes related to extracellular exosomes, specific granules, and endomembrane systems were overrepresented in up-regulated genes specific to severe COVID-19 cases (SetUPCOVID represents the set of up-regulated genes exclusively in COVID-19 patients). **(C)** Genes overexpressed in all severe patients are related to extracellular exosomes, specific granules, and lysosomes (SetUPall represents the set of genes up-regulated in all patients with severe respiratory syndrome, including COVID-19 patients, and other respiratory diseases). **(D)** Protein–protein interaction network formed by differentially expressed gene products. Red nodes represent genes up-regulated only in severe COVID-19 patients (SetUPCOVID), whereas white nodes indicate genes up-regulated in both COVID-19 and non-COVID patients (SetUPall). The top 10 nodes with high degree and betweenness centrality are depicted in the right quadrants, and these nodes (rectangular and labeled nodes) were highlighted in the network. **(E)** Predict transcriptional regulators of up-regulated genes. Bubble size indicates the number of associated targets influenced by the transcription factor. The intensity of the bubble's color corresponds to the NES; NES > 4.0 is considered noteworthy.

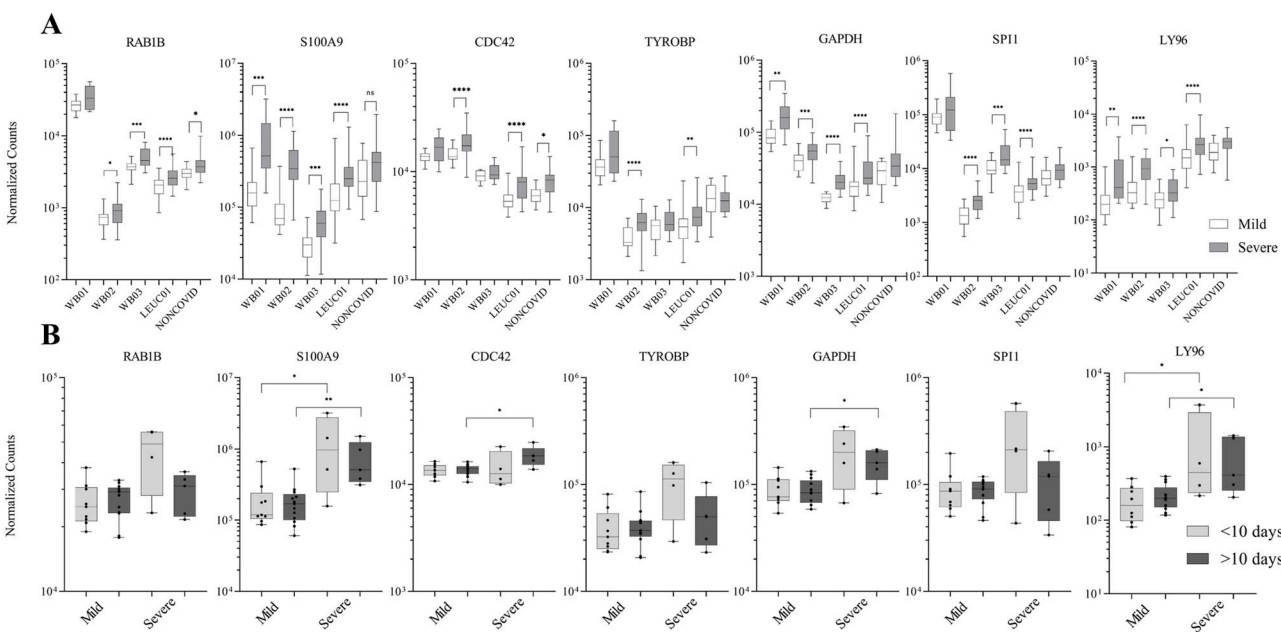

**Figure 4. S100A9 and LY96 expression over time.**
**(A)** Normalized read counts of crucial genes within the protein–protein interaction network. *RAB1B*, *S100A9*, *CDC42*, *TYROBP*, *GAPDH*, *SPI1*, and *LY96* were overexpressed in severe COVID-19 patients (SetUPCOVID). **(B)** Analysis of the WB01 data is divided into two categories: <10 d of symptoms onset and >10 d of symptoms onset until the sample is collected. Data distribution was assessed using the Shapiro–Wilk normality test and the Mann–Whitney test for non-normal data to compute the *P*-value. *P < 0.05, **P < 0.01, ***P < 0.001.

hand, we observed that the "setUPall" showed different enriched binding motif between genes, indicating a transcriptional regulation specific to SARS-CoV-2 infection (Fig 3E).

The landscape of normalized gene expression of central genes in each data is represented in Fig 4. The gene expression of *S100A9*, *GAPDH*, and *LY96* was increased in all data of severe COVID-19 patients. In contrast, *SPI1* and *RAB1B* showed increased expression in three datasets of severe COVID-19 patients. However, the loss of significance in data WB01 could be related to the small number of severe patients in these data (n = 9). To analyze if the differential expression of these genes was related to the time of infection, we searched the metadata for information on the samples collected regarding the moment of onset of symptoms. As a result, we observed that only the data WB01 classified samples according to days of symptoms. Therefore, the WB01 data were subdivided into two groups: sample collected <10 d of symptoms onset and >10 d of symptoms onset (Fig 4B). We observed that the gene expression of *S100A9* and *LY96* was increased in all patients, independent of infection days. However, the expression of *GAPDH* and *CDC42* significantly increased after 10 d of symptom onset.

### Integrated analysis indicated a decrease in the expression of genes related to the T cell receptor complex only in severe COVID-19 patients

The analysis of 179 genes down-regulated in severe COVID-19 data shows that was an enrichment of genes related to the T cell receptor complex and components of nucleoplasm (Fig 5A). The expression of genes such as *TRDV2*, *TRVD1*, *TRAV38-1*, *TRBV3-1*, *TRAV13-1*, and *TRBV2*

was among the 15 most decreased in all data of severe COVID-19. On the other hand, genes down-regulated in all severe patient-coding proteins of MHC class II protein complex, beta-catenin-TCF complex, and components of endoplasmic reticulum membrane (Fig 5B; Table S3). Despite the enrichment of T cell receptor genes in down-regulated genes, the proteins encoded by these genes did not show importance in the interaction network. Instead, TP53, NCL, and NAT10 proteins have the highest network centrality values (Fig 5C). These results suggest the importance of T cell receptor gene expression on the susceptibility of SARS-CoV-2 infection. The search for regulators that could be responsible for a decrease in the expression of these genes resulted in SPI1 (NES = 4.677), MAX (NES = 4.536), and MYC (NES = 4.641). Besides, TP53 (NES = 4.315) also has enriched binding motifs between down-regulated common genes.

The analysis of normalized gene expression of selected genes indicated promising targets for future investigations. The gene expressions *IL2RB*, *TP53*, and *NCL* were significantly decreased in all severe COVID-19 patients (Fig 6A). Furthermore, the significant difference in the decrease of the expression of *IL2RB* and *TP53* occurs mainly after 10 d of the symptom's onset (Fig 6B). However, it is essential to remember that analyzes with a more significant number of participants with classification of time of symptoms are necessary to obtain an expression analysis during disease progression.

### Gene set enrichment analysis (GSEA) indicates genes related to severe COVID-19 phenotype

The data analysis by the GSEA was performed by comparing all patients with severe COVID-19 (n = 139) with other patients,

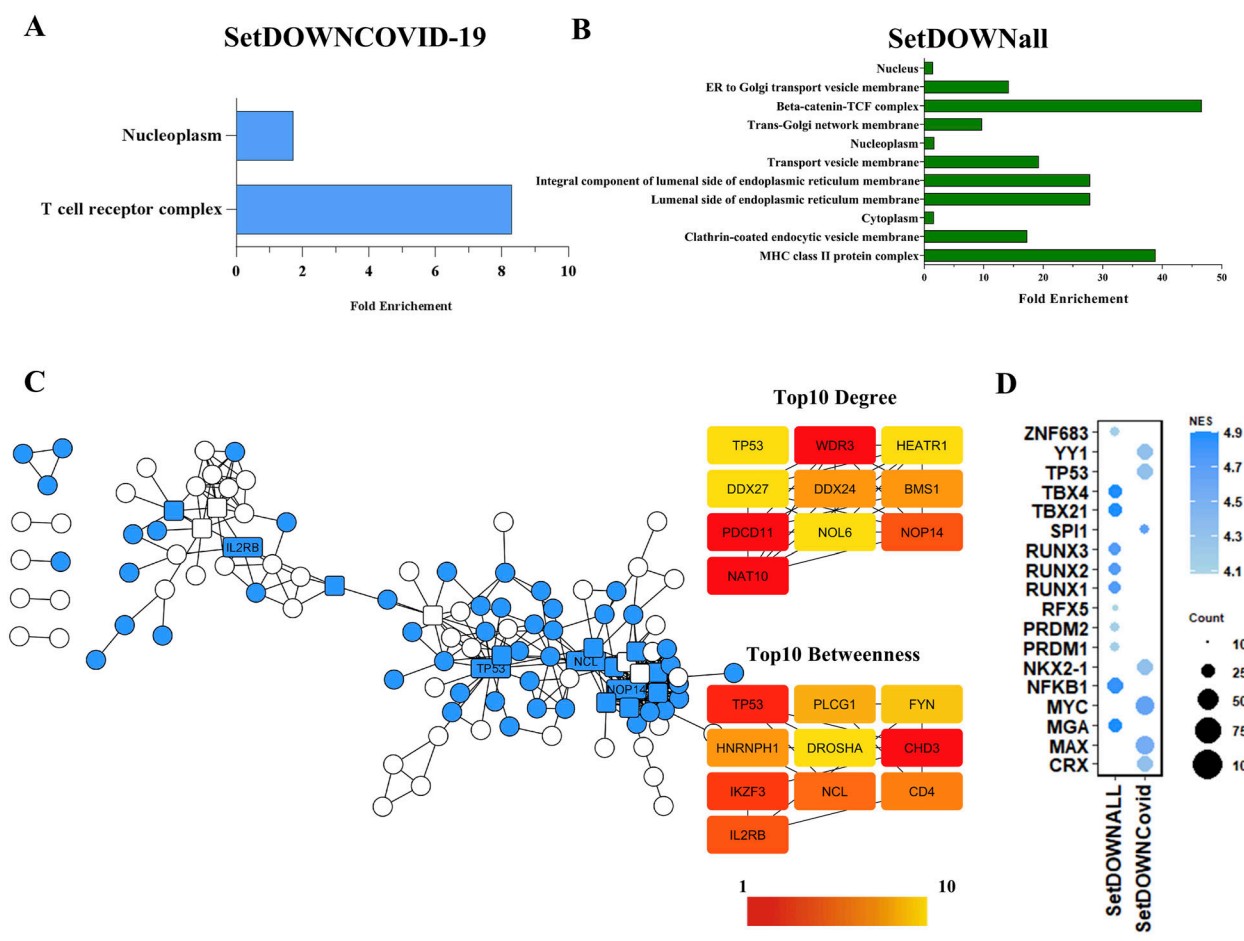

**Figure 5.  Genes related to the T cell receptor complex and nucleoplasm were overrepresented in COVID-19 down-regulated specific genes.**
**(A)** Down-regulated genes in severe patients showed overrepresented genes linked to the T cell receptor complex. The presented components were significant with FDR < 0.05 (SetDOWNCOVID represents the set of genes down-regulated exclusively in COVID-19 patients). **(B)** Genes related to the MHC class II protein and Beta-catenin–TCF complex were overrepresented in genes down-regulated in all severe patients (SetDOWNall represents the set of down-regulated genes in all patients with respiratory syndrome). **(C)** Protein–protein interaction network formed by products of down-regulated DGEs. Blue nodes represent genes exclusively down-regulated in severe COVID-19 patients (SetDOWNCOVID), whereas white nodes represent down-regulated in COVID-19 and non-COVID-19 (SetDOWNall). The top 10 genes with the highest node degree and betweenness centrality were highlighted and labeled. **(D)** Predict transcriptional regulators of down-regulated genes. Bubble size corresponds to the number of targets associated with the transcription factor, whereas the color intensity corresponds to the NES > 4.0, which is considered noteworthy.

including those in the ICU who tested negative for COVID-19 (n = 133). As a result, GSEA identifies the enrichment of genes that codes proteins located in specific granules and endoplasmic reticulum exit sites in severe COVID-19 patients, corroborating with components founded upon the analysis of DEGs by DAVID. On the other hand, cellular components like the T cell receptor complex and MHC protein complex were enriched in a mild phenotype (Table S4). Furthermore, genes related to platelet alpha granule membrane and positive regulation of coagulation were also enriched in severe COVID-19 patients (Table S4). We used the GSEA score to select potential biomarkers with expression related to severe or mild phenotypes. The top 20 genes with the highest score are represented in Fig 7A, and the top five genes have the distribution of expression represented in Fig 7B. *SRPK1, CSGALNACT2, B4GALT5, MEGF9,* and *KLF5* have increased mRNA expression in patients with severe COVID-19 compared with mild and non–COVID-19 patients. The time-dependent analysis did not demonstrate a significant

difference in most genes. However, *SRPK1* expression might be increased after 10 d of symptom onset, and the expression of *KLF5* significantly differed between severe patients in the early and late stages of infection (Fig 7B). In contrast, the expression of genes *CACNA2D3* and *KLRB1* has decreased in severe COVID-19 patients. In contrast, the expression of *FCER1A*, *HLA-DMB*, and *CD1C* was not different between severe COVID-19 and non–COVID-19 patients (Fig 7C).

## Discussion

Here, we employed an integrated and reproducible analysis of RNA-seq whole-blood data published during the COVID-19 pandemic to identify critical molecular targets in the progression of the infection by SARS-CoV-2. The integrated analysis of multiple datasets

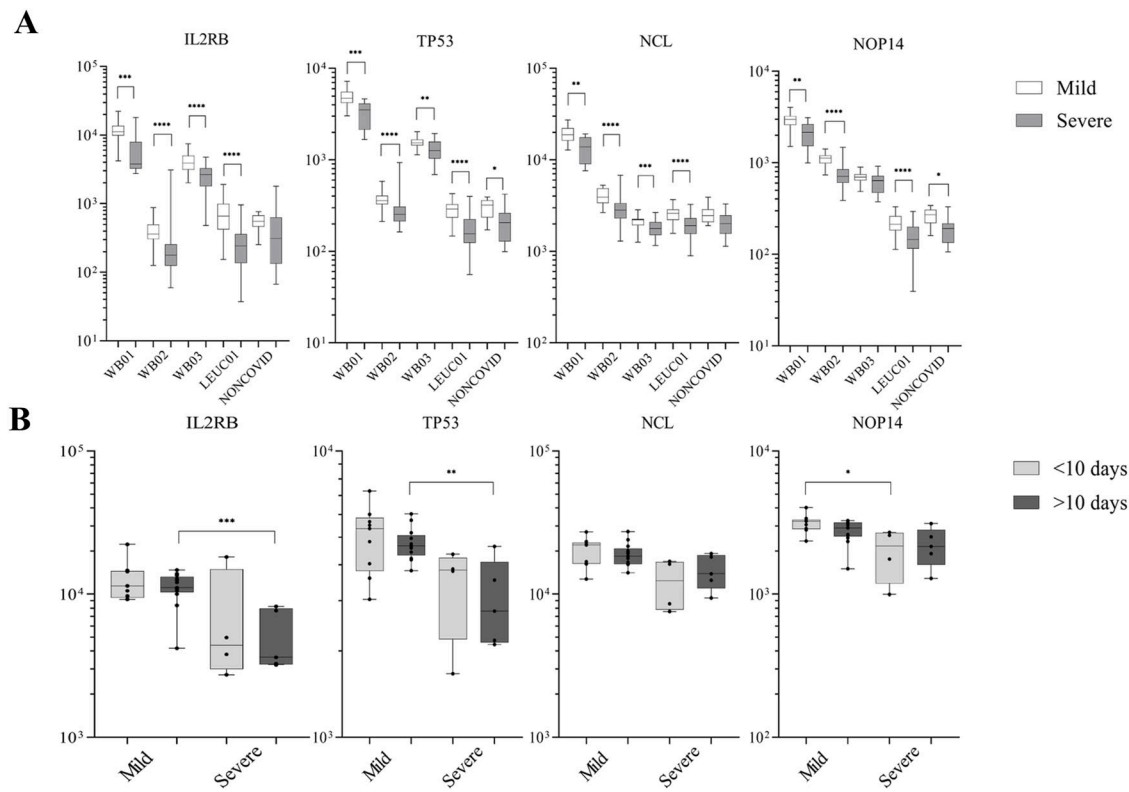

**Figure 6. IL2RB and TP53 expression remain decreased after 10 d of symptom onset in patients with severe COVID-19.**
**(A)** Normalized read counts of important genes to the protein–protein interaction network. *IL2RB*, *TP53*, *NCL*, and *NOP14* were down-regulated in severe COVID-19 patients (SetDOWNCOVID). **(B)** The WB01 dataset was subdivided based on symptom onset (<10 and >10 d) until the sample collection. Data distribution was assessed using the Shapiro–Wilk normality test and the Mann–Whitney test for non-normal data to calculate the *P*-value. *$P < 0.05$, **$P < 0.01$, ***$P < 0.001$.

increases the number of patients evaluated and the genetic variability, including patients from different backgrounds. In this analysis, we highlight the increase of three classes of genes: (1) genes that code proteins located in the endomembrane system (including endoplasmic reticulum exit site until lysosomes), (2) extracellular exosome, and (3) genes related to specific granule and secretory granules (some of them related to neutrophils). The enrichment of these components was also seen in non-COVID-19 patients, showing the importance of this process in severe infections. However, specific genes showed increased expression exclusively in severe COVID-19 patients. SARS-CoV-2 intersects with the endomembrane system in various steps of the viral lifecycle (Sicari et al, 2020; Chen et al, 2022a). The virus–host interactome shows that SARS-CoV-2 proteins interact with many proteins of the host secretory pathway (Gordon et al, 2020; Stukalov et al, 2021). Furthermore, genome-wide CRISPR/Cas9 screens identified host factors in the endomembrane system limiting the infection (Wang et al, 2021a; Daniloski et al, 2021; Schneider et al, 2021). For example, the loss of RAB7A reduces viral entry in the cell by altering endosomal trafficking (Daniloski et al, 2021).

Our work highlighted an increased expression of genes related to the endomembrane system, such as *RAB1B*, *RAB24*, *RAB32*, *JAK3*, and *SELP*, in severe COVID-19 patients' blood samples. *RAB1B* gene encodes a monomeric GTPase that controls ER–Golgi traffic and plays a central role in networks constructed by DEGs, interacting

with many other proteins and connected pathways. Previous studies identified that RAB1B inhibition impairs the maturation of the SARS-CoV-2 spike protein (Veeck et al, 2023). The *CDC42* up-regulated gene also encodes a central protein in PPI. This gene encodes a small GTPase from the Rho family involved in endocytosis (Gundu et al, 2022). We did not find works analyzing the role of CDC42 during a SARS-CoV-2 infection. However, its role in the entry of other RNA viruses has already been evaluated previously (Swaine & Dittmar, 2015).

In addition, lysosomal membrane-related genes also were up-regulated in the blood of severe COVID-19 patients. Differently from other RNA viruses, it has been demonstrated that β-coronaviruses also egress from cells using lysosomal organelles (Ghosh et al, 2020). In this way, the SARS-CoV-2 ORF3a protein facilitates lysosomal transport in a BORC, ARL8b, VAMP7, and STX4-dependent manner (Chen et al, 2021). In this work, we identified an increased expression of lysosomal genes such as *LAMTOR1*, *NEU1*, *GBA1*, *BORCS8*, and *BLOC1S1* in severe COVID-19 patients. The proteins produced by *BORCS8* and *BLOC1S1* are part of the BORC complex and regulate lysosome positioning and movement with ARL8b (Pu et al, 2015). These findings indicate that these genes may be essential in disease promotion because of increased expression in patients with severe COVID-19.

In addition, recent articles have highlighted the potential of extracellular vesicles in promoting virus egress from cells. One of

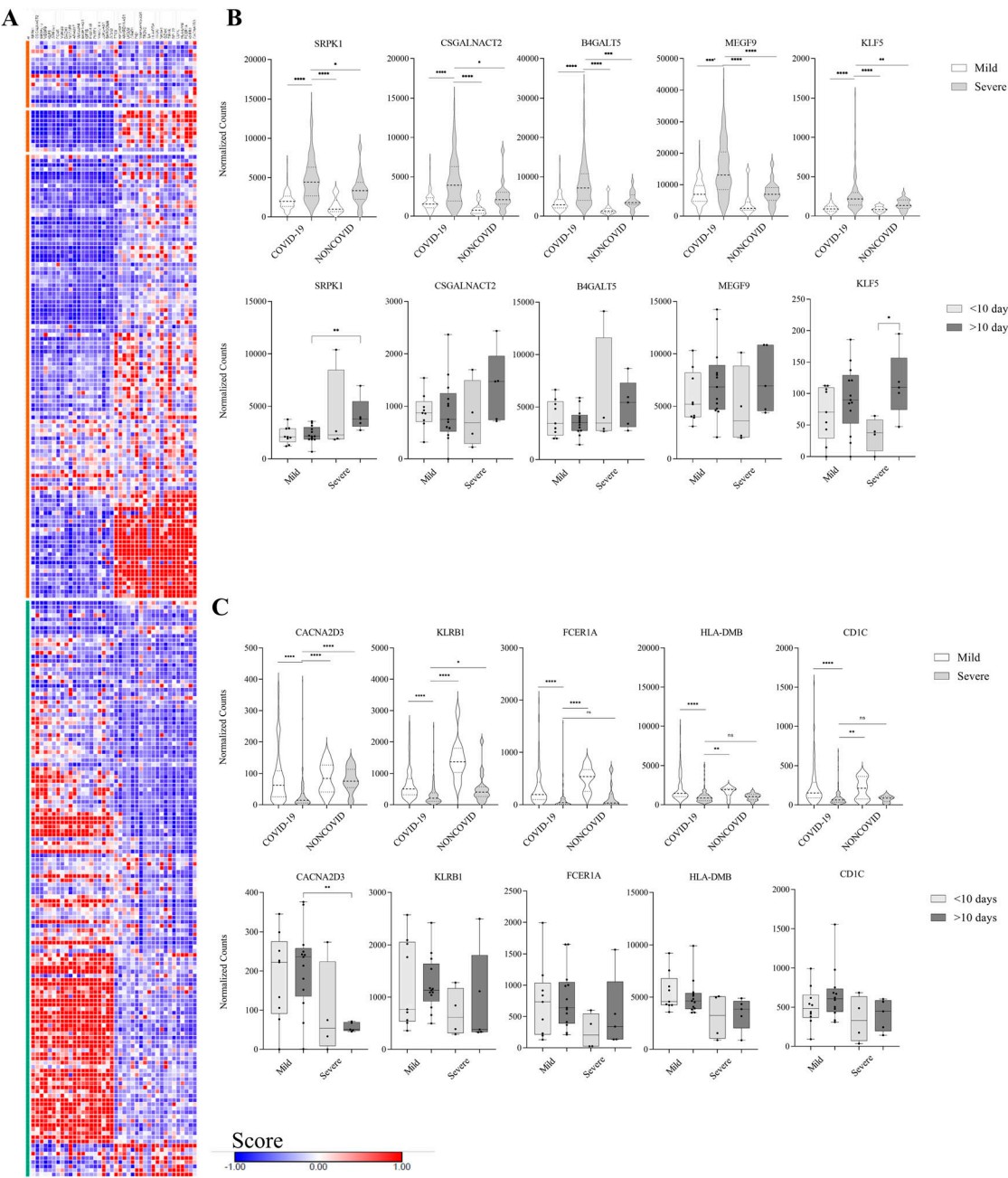

**Figure 7. Gene expression signature analysis of severe COVID-19 phenotype according to gene set enrichment analysis.**
**(A)** Heatmap displaying the top 20 gene scores related to severe COVID-19 phenotype (green band) and the top 20 gene scores related to mild COVID-19 and non-COVID-19 patients (orange band). **(B)** Distribution of normalized gene expression related to severe COVID-19 phenotype. All the genes were significantly overexpressed in severe COVID-19 patients. The lower panel illustrates gene expression based on days of symptom onset in the data WB01 dataset. SRPK1 and KLF5 exhibited overexpression only after 10 d of symptom onset. **(C)** Distribution of normalized gene expression of genes related to mild COVID-19 phenotype. All the genes were significantly down-regulated in severe COVID-19 patients. The lower panel represents gene expression based on days of symptom onset in the WB01 dataset. *CACNA2D3* was the only gene with notable down-regulation expression after 10 d of symptom onset. Data distribution was assessed using the Shapiro–Wilk normality test and the Mann–Whitney test for non-normal data to calculate the *P*-value. *$P < 0.05$, **$P < 0.01$, ***$P < 0.001$.

these articles demonstrated that VeroE6 cells infected with SARS-CoV-2 produce extracellular vesicles larger than exosomes and originate from the Golgi complex. These vesicles contain viral particles that promote virus escape from neutralizing antibodies (Xia et al, 2023). Another study identified that extracellular vesicles

of different sizes were detected in the blood of COVID-19 patients and can be related to the severity of the disease (Lam et al, 2021). Our analysis identified target genes that encode proteins located in exosomes that were crucial to vesicle transport, such as *CD63*, *ATP6V1D*, and *CHMP1A*. CD63, commonly used as an exosome

marker, increased protein expression, and correlated to higher exosome levels in severely ill COVID-19 patients (Aharon et al, 2023). In addition, CD63 expression was increased in plasma cell-free RNA sequences of severe COVID-19 patients (Wang et al, 2022). Corroborating with these data, we also observed an increase in mRNA expression of CD63 in whole blood and leukocytes of severe COVID-19 patients.

The group of genes related to the endomembrane and vesicle is essential at the beginning of the infection for endocytosis, replication, and exit of the virus in the cell. In this work, we explored the hypothesis that individuals with greater expression of these specific genes are susceptible to infection by SARS-CoV-2. However, we acknowledge that our analysis did not verify whether this increased expression also occurs at the primary site of infection. In addition, the expression of these genes in immune cells can contribute to the disease progression. For example, immune cells release exosomes with altered contents during inflammation, and these exosomes play an essential role in mediating inflammation (Chan et al, 2019). Notably, plasma exosomes isolated from COVID-19 patients have already been observed to stimulate cytokine production by PBMC cells from healthy donors (Chen et al, 2022b).

Furthermore, our analysis revealed an increase of granules-located genes, which corroborates the increased expression of genes related to neutrophil activation in datasets WB02, WB03, and LEU01. Neutrophil activation is one of the hallmarks of COVID-19 severity. However, it has been observed that COVID-19 shares neutrophil activation with other inflammatory states (Schimke et al, 2022). This work identified specific overexpressed genes related to neutrophils only in severe COVID-19 patients, such as *SPI1*, *SLPI*, *S100A9*, and *BRI3*, which indicates a specific expression regulation that occurs only in SARS-CoV-2 infection. In this work, the increased expression of *SPI1* and its target was explicitly observed in patients with severe COVID-19, indicating that SPI1 has a vital role in transcriptional regulation during COVID-19. It has been observed that transcription factors like SPI1, which is related to granule–progenitor differentiation, were up-regulated in COVID-19 patients (Wang et al, 2021b). However, Wang and colleagues did not identify significant differences in SPI1 expression between mild and severe patients, which may be because of a limited number of participants in the study (n = 6).

In contrast, our integrated analysis involved multiple datasets, enabling us to identify the specific up-regulation of SPI1 in severe COVID-19 cases. We also identified predicted targets of SPI1 with differential expressions between severe and mild patients. Among these genes, S100A9 has already been suggested as a potential biomarker of COVID-19 severity (Chen et al, 2020a; Silvin et al, 2020). This gene encodes a protein that, together with S100A8, forms an alarmin known as calprotectin (S100A8/A9). This alarmin is released after the activation of neutrophils and binds to TLR4 to promote signaling activation via NF-kB and the secretion of inflammatory cytokines such as IL-6 (Mellett & Khader, 2022). In this work, we showed that the increase of S100A9 occurs not only in the release of alarmin but also in gene expression levels. Furthermore, *S100A9* expression increases in severe COVID-19 patients before 10 d of symptom onset, suggesting that this increase is not just a consequence of prolonged inflammation.

In addition, the *LY96* gene, also known as MD-2, exhibits an expression pattern similar to *S100A9*, suggesting that this gene also could be a good marker of disease severity. MD-2 is an accessory protein required for LPS and HMGB1 inflammatory signaling through TLR4 (Visintin et al, 2006; Yang et al, 2015; Chen et al, 2018; He et al, 2018). However, to the best of our knowledge, there are no articles that associate *LY96* expression with COVID-19 severity.

Among the down-regulated genes, we found enrichment of (1) genes encoding T cell receptor complex proteins only in COVID-9 patients, (2) genes encoding proteins located in the nucleoplasm, and (3) TP53, MYC, and MAX among the master regulators of down-expressed genes. Lymphopenia is a well-known symptom in patients hospitalized for COVID-19. The decrease of T lymphocytes in severe patients may be why we observed a decrease in the expression of specific genes of T cells because the RNA extraction was probably performed with proportionally fewer T cells concerning other leukocytes. However, in addition to genes encoding subunits of the T cell receptor complex, genes related to the T cell-mediated immune response also have decreased expression in patients with severe COVID-19. The *IL2RB* gene, for example, which encodes an IL-2 receptor (IL-2R) chain, emerged as one of the central genes in the PPI network formed by down-regulated COVID-19 gene, indicating that IL2RB may be an essential target for understanding the COVID-19 disease progression and lymphopenia. IL-2R responds to IL-2 cytokine and is expressed in T and NK cells (Fernandez et al, 2019). IL-2 is critical to T cells' proliferation, differentiation, and function (Ross & Cantrell, 2018). A study by Zang and colleagues (2020) observed that the increase of soluble IL-2R is correlated to lymphopenia in COVID-19 patients' plasma (Zhang et al, 2020). Because of the increase in circulating IL-2R, the cellular response to IL2 decreases (Gooding et al, 1995).

Furthermore, plasma IL-2R concentration increased with symptom days (Zhang et al, 2020). Our work further reveals that the expression of IL2RB mRNA is mainly reduced in patients with more than 10 d of symptoms. In addition, GSEA analysis identified a negative relationship between *KLRB1* and *CD1C* expression and the severe phenotype of COVID-19 patients. Both genes are related to NK and T cell antigen presentation. *KLRB1* encodes a transmembrane protein expressed in NK cells and T cells named CD161, and *CD1C* encodes a member of the CD1 family related to MHC proteins (Moody & Cotton, 2017; Wyrożemski & Qiao, 2021).

The second key finding from the integrated analysis of COVID-19 transcriptome data showed decreased expression of nucleoplasm-related genes, including transcription factors, such as *ETS1*, *MYCL*, *NFATC2*, and *TP53*, and genes that encode proteins related to chromatin regulation such as *NCL*, *KAT6B*, and *NAT10*. The *NCL* gene encodes nucleolin, a multifunctional nucleolar protein with many interactions in the network of down-regulated genes. NCL is already related to the entry and replication of several viruses (Jia et al, 2017). Furthermore, it interacts with SARS-CoV-2 proteins, such as N and S genes and Nsp1 and Nsp12 proteins. This interaction was suggested to mediate SARS-CoV-2 induction of p53-dependent apoptosis (Merino et al, 2023). We observed that *NCL* expression was significantly decreased in all datasets from patients with severe COVID-19. In our study, the p53 gene (*TP53*) expression considerably decreased, particularly after 10 d of symptoms. However, we observed that

the expression was similarly diminished in ICU patients who did not have COVID-19.

Furthermore, p53, MYC, and MAX showed binding sites on several down-regulated genes, indicating that these proteins have a potential role in the transcriptional regulation of other genes. P53 is an essential transcriptional regulatory factor well known for its involvement in the cell cycle and apoptosis. However, research has highlighted its role in antiviral immunity, as P53 knockout mice are sensitive to viral infections (Muñoz-Fontela et al, 2005, 2016). In addition, P53 regulates the expression of essential targets for immunity and inflammation, including IRF5 and TLR3 (Mori et al, 2002; Taura et al, 2008). A study by Bordoni et al (2021) observed increased p53 expression in PBMC from COVID-19 patients compared with healthy patients. Therefore, decreased TP53 expression in patients with severe COVID-19 may contribute to the disease's development.

The binding motif for Myc/Max is required for *TP53* transcription (Kirch et al, 1999). Our analysis demonstrates decreased expression of several MYC and MAX targets in patients with severe COVID-19. However, whereas MYC expression was significantly decreased in WB01 and WB03 datasets, *MAX* expression showed no significant change. These findings indicate that the dysregulation of Myc/Max targets is not solely because of the changes in the expression of transcription factors. In other studies, the SARS-CoV-2 Nsp6 protein has already been observed to interact with components of the MGA/MAX complex in fly cardiomyocytes. This interaction diminishes the antagonistic role of this complex in regulating the expression of MYC/MAX targets and increases the expression of genes encoding glycolysis pathways proteins (Zhu et al, 2022). Although we did not observe an enrichment of glycolysis-related DEGs in any gene ontology classifications in blood, it is crucial to consider that *MGA* gene expression was decreased in all severe patients, potentially contributing to the up-regulation of MYC targets in these patients. Nonetheless, MYC has diverse roles in regulating the expression of multiple targets and has many partners, warranting further investigation of its function during the inflammatory process triggered by SARS-CoV-2.

In our study, we have identified potential targets whose expression may be crucial in determining susceptibility to severe forms of COVID-19. Furthermore, assessing their cellular interactions could provide insights into the molecular pathophysiology of SARS-CoV-2 infection. In addition, identifying transcriptional factors that regulate the expression of genes involved in infection and inflammatory response could be pivotal in advancing our understanding of the disease. By comprehending cellular interactions and responses to viruses such as SARS-COV-2, we can better equip ourselves to respond effectively to future pandemics.

# Materials and Methods

### RNA-seq data

RNA-seq datasets of COVID-19 patients were retrieved from the Gene Expression Omnibus public repository and the European Nucleotide Archive. We selected data from blood samples with publicly available raw data and information about disease severity in each patient. We selected three datasets of whole-blood samples RNA-seq submitted under the accession number GSE161731 (McClain et al, 2021), GSE172114 (Carapito et al, 2022), and PRJEB47548 (Jackson et al, 2022), and one RNA-seq of leukocytes from blood submitted under the accession number GSE157103 (Overmyer et al, 2021). For this work, each selected data were renamed as WB01 (GSE161731), WB02 (GSE172114), LEU_01 and NONCOVID01 (GSE157103), and WB03 (PRJEB47548). Table S5 includes all available information regarding disease severity, age, and gender. The analysis of DEGs below was performed by comparing data within the same project.

### Data processing

The raw datasets were uploaded in usegalaxy.eu web platform (Galaxy Community et al, 2022) to reanalyze the data in an accessible way. The data quality was analyzed using FastQC (Galaxy Version 0.73+galaxy0) and MultiQC (Galaxy Version 1.11+galaxy1) tools (Ewels et al, 2016). The RNA-seq aligner STAR (Galaxy Version 2.7.10b+galaxy3) (Dobin et al, 2013) was used to align all data with the reference genome GRCh38 and featureCounts (Galaxy Version 2.0.3+galaxy1) (Liao et al, 2014) was used to count align reads. The DEseq2 (Galaxy Version 2.11.40.7+galaxy2) (Love et al, 2014) was applied to compare severe and mild COVID-19 patients and to calculate the normalized count expression.

### Gene ontology and regulators analysis

We analyzed the gene ontology (GO) classification of the (DEGs) using DAVID tools (Huang et al, 2009). We classified the genes in line with biological processes and cellular components that belong. Only the process with FDR < 0.05 was considered for analysis. To predict the master regulators, the DEGs were uploaded in Cytoscape v.3.8.2 (Shannon et al, 2003). We used the plugging iRegulon v.1.3 (Janky et al, 2014) to predict regulators of DEGs. The genes were separated into up-regulated and down-regulated genes, and we searched by transcription factors and ranked by putative regulatory region in 20 kb centered around the transcription start site. Only regulators with normalized enrichment scores (NES) > 4.0 were analyzed.

### PPI

We used the String-DB v. 11.5 (Szklarczyk et al, 2023) web platform to construct a protein–protein interaction network and Cytoscape v3.8.2 to analyze the network. The centrality scores degree and betweenness were calculated using CentiScaPe v. 2.2 (Scardoni et al, 2014).

### GSEA

Normalized count tables were analyzed in the GSEA desktop application (version 4.3.2) using 1,000 gene set permutations (Subramanian et al, 2005). Phenotype labels were divided into COVID-19 patients (severe and mild) and non-COVID-19 patients (severe and mild). The severe COVID-19 patients (139 samples) were compared with the others (133 samples), aiming to calculate the score of genes. The genes were ranked using the Signal2Noise

metric and classified according to GO datasets from MsigDB (c5.go.bp.v2023.1 and c5.go.cc.v2023.1).

## Statistical analysis

The statistical analysis was performed by bioinformatics tools as indicated in the respective topic. We executed the Shapiro–Wilk normality test to compare normalized expressions between two conditions to analyze the data distribution. Then, Mann–Whitney test for non-normal data was used in GraphPad Prism 8 to calculate the *P*-value. *P*-values < 0.05 were considered significant.

# Supplementary Information

# Acknowledgements

This work was supported by the Conselho Nacional de Desenvolvimento Cientfico e Tecnológico (CNPq, Brazil) and Coordenação de Aperfeiçoamento de Pessoal de Nível Superior (CAPES, Brazil).

## Author Contributions

TT Oliveira: conceptualization, data curation, formal analysis, investigation, methodology, and writing—original draft.

JF Freitas: conceptualization, data curation, formal analysis, methodology, and writing—review and editing.

VPB de Medeiros: conceptualization, formal analysis, methodology, and writing—review and editing.

TJdS Xavier: conceptualization, investigation, methodology, and writing—review and editing.

LF Agnez-Lima: conceptualization, data curation, formal analysis, funding acquisition, investigation, methodology, project administration, and writing—review and editing.

## Conflict of Interest Statement

The authors declare that they have no conflict of interest.

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
