## [Reviewer comments · Life Science Alliance]

Life Science Alliance

Integrated analysis of RNA-seq datasets reveals novel targets and regulators of disease severity

Thais T Oliveira, Julia F Freitas, Viviane P B Medeiros, Thiago J S Xavier and Lucymara Fassarella Agnez-Lima
DOI: DOI: <https://doi.org/10.26508/lsa.202302358>

Corresponding author(s): Lucymara Fassarella Agnez-Lima (Federal University of Rio Grande do Norte)

Review Timeline:

Submission Date:	2023-09-06
Editorial Decision:	2023-11-06
Revision Received:	2023-11-29
Editorial Decision:	2024-01-11
Revision Received:	2024-01-12
Accepted:	2024-01-16

Transaction Report:

November 6, 2023

Re: Life Science Alliance manuscript #LSA-2023-02358-T

Lucymara Fassarella Agnez-Lima
Universidade Federal do Rio Grande do Norte

Dear Dr. Agnez-Lima,

Thank you for submitting your manuscript entitled "Unveiling molecular signatures of COVID-19 progression: integrated analysis of RNA-seq datasets reveals novel targets and transcriptional regulators associated with disease severity." to Life Science Alliance. The manuscript was assessed by expert reviewers, whose comments are appended to this letter. We invite you to submit a revised manuscript addressing the Reviewer comments.

Thank you for this interesting contribution to Life Science Alliance. We are looking forward to receiving your revised manuscript.

Sincerely,

B. MANUSCRIPT ORGANIZATION AND FORMATTING:

Reviewer #1 (Comments to the Authors (Required)):

Short summary: Oliveira TT et al. re-analyzed four publicly blood transcriptomic datasets, identified genes associated with COVID-19 disease severity, characterized the pathways those genes were part of, and combined using protein-protein interaction data.

The study is highly collaborative, with no experimental validation of the findings and no attempt to confirm the identified list of genes on RNA-Seq data of the respiratory system or even in vitro transcriptomic datasets.

Main points: Data collection and description/ what criteria were used to select only those four datasets?

page 4/ "optimal environment" define what you mean by optimal environment

page 7/ why combining moderate/severe participants instead of investigating them separately

Figure 1/What does "MV*" refer to?

page 9/ "enriched binding motifs between upregulated genes" do you mean "enriched binding motifs in the regulatory regions of upregulated genes"

minor issues: abstract/ replace "genes coding proteins" by "genes coding for proteins".

page 4/ "mild, moderate,... (ARDS)" rephrase to make clear that critical COVID-19 are marked by acute respiratory failure and ARDS.

page 4/Is it "gender" or "sex"?

Reviewer #2 (Comments to the Authors (Required)):

Comments for the Author :

This manuscript identified key genes and master regulators of these gene sets, which may play an essential role in the cellular response to infection.

Below are my major concerns:

(1) This study combined moderate and severe patients into the severe group, as both were hospitalized. However, in COVID-19 data, "Moderate" and "Severe" are typically two distinct clinical classifications with differences between them. In light of the substantial amount of COVID-19 data available, might it be more suitable to classify patients into three categories: mild, moderate, and severe?

(2) I primarily focus on the findings such as "Integrated analysis shows overexpression of genes related to exosome, lysosomal membrane, and specific granules only in severe COVID-19 patients," and "Integrated analysis indicated a decrease in the expression of genes related to the T cell receptor complex only in severe COVID-19 patients." However, it seems that these analyses mainly involve gene enrichment and network analysis, and perhaps further downstream analysis or validation should be considered.

(3) The author noted that certain genes exhibited observed differences in the severity of this disease when compared to other flu and respiratory diseases, which is interesting. However, I couldn't find the corresponding content.

Answer to Reviewer #1:

First, we thank the referee for their constructive comments, as we believe they have improved our manuscript. Below are the point-by-point answers:

Short summary: Oliveira TT et al. re-analyzed four publicly blood transcriptomic datasets, identified genes associated with COVID-19 disease severity, characterized the pathways those genes were part of, and combined using protein-protein interaction data. The study is highly collaborative, with no experimental validation of the findings and no attempt to confirm the identified list of genes on RNA-Seq data of the respiratory system or even in vitro transcriptomic datasets.

Main points: Data collection and description/ what criteria were used to select only those four datasets?

Answer: The data selection process relied on the availability of raw sequencing files in public databases and the classification of samples according to disease severity. In addition, data from blood samples were used to assess immune cell gene expression. Unfortunately, many public data lack information, and some projects do not make the raw data available.

page 4/“optimal environment” define what you mean by optimal environment

Answer: We agree with the referee and correct the text to make it clearer. The optimal environment refers to the cellular apparatus for virus replication, assembly, and egress from cells.

page 7/ why combining moderate/severe participants instead of investigating them separately?

Answer: The standardizing of patient divisions was a challenge in analyzing several datasets from diverse sources, given the variations in how the researchers categorized their patients according to distinct criteria. Therefore, we merged the data from WB03 samples (E-MTAB-10926) into two groups since it was the only dataset with distinctions between mild, moderate, and severe patients. This decision aimed to align group divisions as closely as possible for a comparative analysis. In this context, the moderate group comprised patients requiring oxygen assistance during hospitalization. In addition, the PCA analysis showed more similarities between the moderate and severe groups than the mild group. Therefore, we chose to combine patients with moderate and severe symptoms in the same group.

Figure 1/What does “MV*” refer to?

Answer: The abbreviation *ICU-MV means intensive care unit patients requiring mechanical ventilation. This information was added to the figure caption.

page 9/“enriched binding motifs between upregulated genes” do you mean “enriched binding motifs in the regulatory regions of upregulated genes”

Answer: Yes, thanks for the correction. The term was corrected in the manuscript (page 9).

minor issues: abstract/ replace “genes coding proteins” by “genes coding for proteins”.

Answer: We agree with the referee; the term was corrected.

page 4/“mild, moderate,... (ARDS)” rephrase to make clear that critical COVID-19 are marked by acute respiratory failure and ARDS.

Answer: The sentence was corrected as suggested.

page 4/Is it “gender” or “sex”?

Answer: In this case, the correct term is “sex”. The term was changed in the manuscript.

Answer to Reviewer #2

First, we thank the referee for their constructive comments, as we believe they have improved our manuscript. Below are the point-by-point answers:

Comments for the Author :

This manuscript identified key genes and master regulators of these gene sets, which may play an essential role in the cellular response to infection.

Below are my major concerns:

(1) This study combined moderate and severe patients into the severe group, as both were hospitalized. However, in COVID-19 data, “Moderate” and “Severe” are typically two distinct clinical classifications with differences between them. In light of the substantial amount of COVID-19 data available, might it be more suitable to classify patients into three categories: mild, moderate, and severe?

Answer: The standardizing of patient divisions was a challenge in analyzing several datasets from diverse sources, given the variations in how the researchers categorized their patients according to distinct criteria. Therefore, we merged the data from WB03 samples (E-MTAB-10926) into two groups since it was the only dataset with distinctions between mild, moderate, and severe patients. This decision aimed to align group divisions as closely as possible for a comparative analysis. In this context, the moderate group comprised patients requiring oxygen assistance during hospitalization. In addition, the PCA analysis showed more similarities between the moderate and severe groups than the mild group. Therefore, we chose to combine patients with moderate and severe symptoms in the same group.

(2) I primarily focus on the findings such as “Integrated analysis shows overexpression of genes related to exosome, lysosomal membrane, and specific granules only in severe COVID-19 patients,” and “Integrated analysis indicated a decrease in the expression of genes related to the T cell receptor complex only in severe COVID-19 patients.” However, it seems that these analyses mainly involve gene enrichment and network analysis, and perhaps further downstream analysis or validation should be considered.

Answer: Thanks for your comment. We acknowledge that the absence of validation of the findings is a limitation in studies that utilize public data due to limitations in accessing the original biological samples. However, our team is actively seeking resources for sequencing local patients and conducting additional analyses to validate the outcomes of this study. In addition, exploring the previously generated data from a new perspective is valuable to the scientific community, given the abundance of unexplored data available. Moreover, the insights from this analysis can guide other research teams, contributing to broader scientific understanding.

(3) The author noted that certain genes exhibited observed differences in the severity of this disease when compared to other flu and respiratory diseases, which is interesting. However, I couldn’t find the corresponding content.

Answer: We categorized differentially expressed genes into four groups: SetUPCOVID, represents the set of upregulated genes exclusively in COVID-19 patients; SetDOWNCOVID represents the set of genes downregulated exclusively in COVID-19 patients; SetUPall represents the set of genes upregulated in all patients with severe respiratory syndrome, including COVID-19 patients and other respiratory diseases; and SetDOWNall represents the set of downregulated genes in all patients with respiratory syndromes. Figures 3 to 6 show the differences between these groups. We observed a set of genes differentially expressed exclusively in COVID-19 patients. We included the description of these groups in the figure legends. Thank you for this comment.

January 11, 2024

RE: Life Science Alliance Manuscript #LSA-2023-02358-TR

Dr. Lucymara Fassarella Agnez-Lima
Federal University of Rio Grande do Norte
Biologia Celular e Genética
Centro de Biociências
Natal, RN 59058900
Brazil

Dear Dr. Agnez-Lima,

Thank you for submitting your revised manuscript entitled "Integrated analysis of RNA-seq datasets reveals novel targets and regulators of disease severity". We would be happy to publish your paper in Life Science Alliance pending final revisions necessary to meet our formatting guidelines.

- please be sure that the authorship listing and order is correct
- please add a Category for your manuscript in our system

A. FINAL FILES:

B. MANUSCRIPT ORGANIZATION AND FORMATTING:

****It is Life Science Alliance policy that if requested, original data images must be made available to the editors. Failure to provide**

original images upon request will result in unavoidable delays in publication. Please ensure that you have access to all original data images prior to final submission.**

The license to publish form must be signed before your manuscript can be sent to production. A link to the electronic license to publish form will be available to the corresponding author only. Please take a moment to check your funder requirements.

Sincerely,

Reviewer #1 (Comments to the Authors (Required)):

Oliveira TT et al. addressed all my edit suggestions/answered my concerns in this new manuscript version.

January 16, 2024

RE: Life Science Alliance Manuscript #LSA-2023-02358-TRR

Dr. Lucymara Fassarella Agnez-Lima
Federal University of Rio Grande do Norte
Biologia Celular e Genética
Centro de Biociências
Natal, RN 59058900
Brazil

Dear Dr. Agnez-Lima,

Thank you for submitting your Research Article entitled "Integrated analysis of RNA-seq datasets reveals novel targets and regulators of disease severity". It is a pleasure to let you know that your manuscript is now accepted for publication in Life Science Alliance. Congratulations on this interesting work.

DISTRIBUTION OF MATERIALS:

Again, congratulations on a very nice paper. I hope you found the review process to be constructive and are pleased with how the manuscript was handled editorially. We look forward to future exciting submissions from your lab.

Sincerely,
